# Hearing Function: Identification of New Candidate Genes Further Explaining the Complexity of This Sensory Ability

**DOI:** 10.3390/genes12081228

**Published:** 2021-08-10

**Authors:** Maria Pina Concas, Anna Morgan, Fabrizio Serra, Andries Paul Nagtegaal, Berthe C. Oosterloo, Sudha Seshadri, Nancy Heard-Costa, Guy Van Camp, Erik Fransen, Margherita Francescatto, Giancarlo Logroscino, Rodolfo Sardone, Nicola Quaranta, Paolo Gasparini, Giorgia Girotto

**Affiliations:** 1Institute for Maternal and Child Health—IRCCS, Burlo Garofolo, 34127 Trieste, Italy; anna.morgan@burlo.trieste.it (A.M.); fabrizio.serra@burlo.trieste.it (F.S.); paolo.gasparini@burlo.trieste.it (P.G.); giorgia.girotto@burlo.trieste.it (G.G.); 2Department of Otorhinolaryngology, Erasmus Medical Center, 3015 CE Rotterdam, The Netherlands; a.nagtegaal@erasmusmc.nl (A.P.N.); b.oosterloo@erasmusmc.nl (B.C.O.); 3Department of Epidemiology, Erasmus Medical Center, 3015 CE Rotterdam, The Netherlands; 4Framingham Heart Study, Framingham, MA 01702, USA; Seshadri@uthscsa.edu (S.S.); nheard@bu.edu (N.H.-C.); 5Glenn Biggs Institute for Alzheimer’s and Neurodegenerative Diseases, University of Texas Health Sciences Center, San Antonio, TX 78229, USA; 6Department of Neurology, Boston University School of Medicine, Boston, MA 02118, USA; 7Center of Medical Genetics, University of Antwerp and Antwerp University Hospital, 2650 Antwerp, Belgium; guy.vancamp@uantwerpen.be; 8Department of Biomedical Sciences, University of Antwerp, 2650 Antwerp, Belgium; erik.fransen@uantwerpen.be; 9Department of Medicine, Surgery and Health Sciences, University of Trieste, 34139 Trieste, Italy; margherita.francescatto@burlo.trieste.it; 10Department of Basic Medical Sciences, Neuroscience and Sense Organs, University of Bari “Aldo Moro”, 70121 Bari, Italy; giancarlo-logroscino@uniba.it; 11Population Health Research Unit, National Institute of Gastroenterology and Research Hospital IRCCS S. De Bellis, Castellana Grotte, 70013 Bari, Italy; rodolfo.sardone@uniba.it; 12Otolaryngology Unit, Department of Basic Medical Science, Neuroscience and Sense Organs, University of Bari Aldo Moro, 70121 Bari, Italy; nicolaantonioadolfo.quaranta@uniba.it

**Keywords:** hearing, GWAS, meta-analysis, inner ear, gene expression

## Abstract

To date, the knowledge of the genetic determinants behind the modulation of hearing ability is relatively limited. To investigate this trait, we performed Genome-Wide Association Study (GWAS) meta-analysis using genotype and audiometric data (hearing thresholds at 0.25, 0.5, 1, 2, 4, and 8 kHz, and pure-tone averages of thresholds at low, medium, and high frequencies) collected in nine cohorts from Europe, South-Eastern USA, Caucasus, and Central Asia, for an overall number of ~9000 subjects. Three hundred seventy-five genes across all nine analyses were tagged by single nucleotide polymorphisms (SNPs) reaching a suggestive *p*-value (*p* < 10^−5^). Amongst these, 15 were successfully replicated using a gene-based approach in the independent Italian Salus in the Apulia cohort (*n* = 1774) at the nominal significance threshold (*p* < 0.05). In addition, the expression level of the replicated genes was assessed in published human and mouse inner ear datasets. Considering expression patterns in humans and mice, eleven genes were considered particularly promising candidates for the hearing function: *BNIP3L*, *ELP5*, *MAP3K20*, *MATN2*, *MTMR7, MYO1E*, *PCNT*, *R3HDM1*, *SLC9A9, TGFB2,* and *YTHDC2*. These findings represent a further contribution to our understanding of the genetic basis of hearing function and its related diseases.

## 1. Introduction

One of the most complex mechanisms in humans is the sense of hearing, which has become a cornerstone of our communication, integration, and sociality [1]. At the base of this fundamental ability is the auditory system, an intricate apparatus aimed at converting mechanical soundwaves into electrical impulses that the brain can process [1]. The key point of this change is performed through sensory hair cells by the organ of Corti, a specialized epithelium contained in the cochlea, a fluid-filled coiled structure located in the inner ear.

Understanding the biology behind the hearing function and the genes/proteins involved has always been complicated. In fact, due to its location, it is possible to take tissue samples from the auditory system in small quantities and only in particular cases. Therefore, biochemical approaches similar to those used to study the visual and olfactory system have limited effectiveness. On the other hand, genetic approaches, such as next-generation sequencing of patients affected by hearing loss or Genome-Wide Association Studies (GWAS) in large cohorts of individuals, have proven useful, revealing that in such a complex mechanism, genetic factors play a fundamental role. In addition, several studies on heritability estimates of hearing impairment have been performed, showing a range between 25% and 75% in different populations [2,3,4,5,6]. To date, more than 170 loci and 123 genes have been reported as involved in hereditary non-syndromic hearing impairment (http://hereditaryhearingloss.org, accessed on 7 May 2021). As regards hearing function and loss (e.g., Age-Related Hearing Loss) some candidate genes have been successfully detected through several GWAS studies. Some examples are *CDH13*, *GRM8*, *ANK2*, *SLC16A6*, *ARSG*, *RIMBP2*, *DCLK1* [7], *SIK3* [8], *PCDH20,* and *SLC28A3* [9] for hearing function, while *SPIRE2*, *ILDR1,* and *ISG20* [10] have been detected for Age-Related Hearing Loss.

However, despite the efforts made so far, the genetic mechanisms underlying deafness are still not fully understood, and even less is known about the genes that play a role in modulating hearing function. Hence, to investigate this trait, a precise evaluation of the involved subjects’ hearing ability and careful clinical characterization is essential. We present the results of a multistep approach made possible thanks to the availability of this information and the matching of genetic data from ten different cohorts from Northern and Southern Europe, the South-Eastern USA, the Caucasus, and Central Asia. This approach was based on (1) a large GWAS discovery meta-analysis involving more than 9000 individuals; (2) the replication in an independent cohort of 1774 subjects from Southern Italy; (3) the evaluation of the expression of the replicated genes datasets describing expression levels in the human and mouse inner ear.

## 2. Materials and Methods

### 2.1. Involved Cohorts

Ten cohorts with audiometric phenotypes were included in this work: four belonged to the G-EAR Consortium (INGI-Friuli Venezia Giulia (FVG) [11], INGI-Val Borbera (VBI) [11], INGI-Carlantino (CAR) [11], and Silk Road (SR) [12]), one to the Age-Related Hearing Impairment Study Antwerp cohort (AWP) [13], three more to the Rotterdam Study (RS1, RS2, and RS3) [14,15], one to the Framingham Heart Study cohort (FHS) [16], and the last, used as a replica, was the Salus in Apulia Study cohort (SIA) [17]. All studies to which the different cohorts belong have received the ethical approval of the respective promoting Institutions; all participating subjects signed informed consent at the time of recruitment. A brief description of each study can be found in the Appendix A.

FVG, VBI, CAR, SR, AWP, RS1, RS2, RS3, and FHS were combined for the discovery meta-analysis, reaching an overall sample of 9057 subjects; for replication in an independent sample, we used the 1774 subjects of SIA.

### 2.2. Analysed Phenotypes

For each of the cohorts involved in the discovery phase, the hearing thresholds of each participant at 250 Hz, 500 Hz, 1 kHz, 2 kHz, 4 kHz, and 8 kHz for both ears were available, as part of pure-tone audiometry assessment performed by each centre (Appendix A). The hearing function was described as a set of nine quantitative traits, as previously introduced in the literature [18]: the thresholds at the six frequencies listed above and three pure-tone averages (PTAs) at low, medium, and high frequencies (respectively, PTAL: 0.25, 0.5, 1 kHz; PTAM: 0.5, 1, 2 kHz; PTAH: 4, 8 kHz), which were prepared following a previously described protocol [18]. The same procedure was used in the replication cohort, SIA, for which only data for PTAM and PTAL were available.

To avoid non-genetic variations in the hearing phenotype, the best hearing ear was chosen and all the individuals displaying unilateral hearing loss have been excluded. Individuals suffering from any forms of inherited hearing loss, exposed to noise or ototoxic medications, as well as subjects affected by diabetes or other systemic diseases potentially leading to hearing loss defects, were removed from the study.

### 2.3. Genotyping, Quality Control, and Imputation

Detailed information about the genotyping of each cohort is reported in Appendix A. Briefly, DNA was extracted from blood or saliva samples and genotyped with different platforms. Standard quality control was performed in each cohort separately. Further inclusion/exclusion criteria were applied in the specific studies. Genotypes were referred to the forward strand and reported with the coordinates of the 1000 Genomes Project build 37 [19]. Individual studies imputed genotype data to HRC [20], 1000 Genomes, or denser reference panels [11] with different imputation tools.

### 2.4. GWAS Analysis and Meta-Analysis

Genome-wide association studies for the nine traits indicated above were carried out employing linear mixed model regression, assuming the additive genetic model and assessing genotype–phenotype association through Wald’s test. Since the traits were already adjusted for gender and age, only genomic kinship (FVG, VBI, CAR, SR, AWP) or five PCs (RS1, RS2, RS3, FHS) were added as covariates. FVG, VBI, CAR, SR, and AWP were analyzed in R (www.r-project.org, accessed on 7 May 2021), using the GRAMMAR-γ method (as implemented in GenABEL [21] on genotyped variants and MixABEL [22] on the imputed ones). We performed association analyses concerning RS1, RS2, RS3, and FHS using RVTESTS v20171009 [23]. Association results were checked for evidence of inflation; variants with minor allele frequency <1% and imputation quality score <0.4 were removed. Summary statistics from individual cohorts were meta-analyzed in METAL v2011-03-25 [24], applying METAL’s STDERR scheme and performing genomic control. Genome-wide significance was set to 5 × 10^−8^: as the traits we analyzed were not independent, we did not conduct any further correction of the significance threshold for multiple testing (see [7,9]); an association was considered suggestive if it presented a *p* < 10^−5^. To ensure the robustness of our work, for the replication step, we only considered variants found in more than 50% of the sample size (i.e., *n* = 4595) with a *p*-value < 1 × 10^−5^. SNPs were annotated with the Variant Effect Predictor tool (VEP, https://www.ensembl.org/info/docs/tools/vep/index.html, accessed on 7 May 2021) [25] to determine their distance from the closest genes and to obtain functional characteristics (i.e., whether they were contained within an intronic, exonic, or intergenic region). The intergenic variants were mapped to the nearest gene within a 250 kb range. The list of the genes detected in this set of variants was used for the replication study. Long non-coding RNAs (LINC) genes or genes with unknown function identified with LOC and FAM symbols or pseudogenes were excluded.

### 2.5. Replication of the Detected Genes in an Independent Cohort

SNP-based association test for SIA on the available two phenotypes (PTAM and PTAH) were performed in GEMMA v0.98 [26], and the results were used for a gene-based association test performed with MAGMA v1.07 [27] through the online platform of FUMA-GWAS [28].

Replication of the genes detected in the meta-analyses was sought in both PTAM and PTAH. As commonly accepted for replication studies, a 5% significance threshold was considered for this phase.

### 2.6. Gene Expression in the Inner Ear

For each replicated gene, we extracted expression data from the human inner ear transcriptome published in [29]. Briefly, the dataset reports normalized (fragment per kilobase per million, FPKM) expression values, recorded in six human samples (three cochleae, one ampulla, one vestibule and one saccule). We additionally identified mouse orthologs for each of the replicated genes and extracted their microarray expression values from [30]. The dataset includes quantile normalized microarray expression data profiled in hair cells (HC), epithelial non-hair cells (ENHC), and non-epithelial cells (NEC) from either cochlea (c) or vestibule (v) epithelia in triplicates. All the plots were created using the ggplot2 R package [31].

## 3. Results

In this work, we used audiometric and genetic data from ten different cohorts to investigate the genetic determinants of hearing function through a multistep approach. First, in the discovery phase, we performed a GWAS meta-analysis from nine cohorts in a sample set of 9000 subjects for each trait. Subsequently, a replication phase was carried out using a gene-based approach; in the final phase, we used a published RNA-seq human inner ear dataset [29] and mouse microarray expression data [30] to assess the expression levels of the genes of interest identified in the previous steps. Figure 1 summarizes the workflow of the study.

Manhattan and QQ plots of all the nine GWAS meta-analyses are shown in Appendix A. A total of 966 associations at a *p*-value < 1 × 10^−5^ between SNPs and audiometric traits were detected, identifying 881 unique SNPs spanning 375 different genes (Appendix A). Their VEP annotation is available in Appendix A.

Fifteen genes out of 375 were replicated in the SIA cohort, as displayed in Table 1 and Appendix A.

The most significantly associated genes in the discovery sample are *MTNR7* and *BNIP3L*. The first was associated with 1 kHZ (rs12155974, *p*-value 2.35 × 10^−7^) and the second with 8 kHZ (rs17402216, *p*-value 4.69 × 10^−7^). Both genes were replicated either in PTAM and PTAH phenotypes. Two genes, *ELP5* and *SLC9A9*, were associated in the discovery sample with more than one trait: *ELP5* with 4 kHZ and PTAH and *SLC9A9* with 250 HZ and PTAL. Both were replicated in the PTAM trait. Figure 2 highlights the results for each studied trait.

The 15 replicated genes were either on different chromosomes or far apart (>10 Mb) and could therefore be considered independent loci.

The expression patterns of these 15 genes in the human inner ear were investigated using RNA-seq expression data from [29] and are shown in Figure 3A. The corresponding FPKM values are reported in full in Appendix A.

With the exception of *OOSP1* (data not available), *CDC177* (not expressed), and *DLGAP2* (weakly expressed in only one tissue), all the genes had at least weak expression in all the tissues profiled. In particular, seven genes (*BNIP3L, MAP3K20*, *MATN2*, *MYO1E*, *R3HDM1, TGFB2,* and *YTHDC2*) had medium expression in all tissues. Between these, *MATN2* showed strong expression in the cochlea, while *TGFB2* and *BNIP3L* showed strong expression in saccule.

Figure 3B shows the expression profiles of the mouse orthologs of the 15 replicated genes in the microarray dataset published in [29]. All expression values are reported in Appendix A. Paralleling to some extent what was observed in human, eight of the genes displayed (in at least one probe) at least medium expression in all the cell types profiled: *Bnip3l*, *Elp5*, *Matn2*, *Myo1e*, *Pcnt*, *Ptprn2*, *R3hdm1,* and *Tgfb2*. Differential expression analysis from [29] indicates that the genes *Mtmr7*, *R3hdm1* (probe 2091509), and *Slc9a9* are characterized by higher expression in the HC with enrichment in cochlear HC (Figure 3C). Mouse expression data were not available for *Cdc177*, *Dlgap2,* and *Oosp1.*

## 4. Discussion

In this work, we performed the most comprehensive GWAS meta-analysis to study the hearing function, with over 9000 subjects involved in the discovery phase and a further 1774 in the replication phase. This large sample size allowed us to identify a relatively high number of genomic loci associated with one or more hearing phenotypes. Indeed, our meta-analysis led to the identification of 15 candidate genes. Among them, eleven genes appeared particularly interesting, considering their expression pattern in the human and mouse inner ear, and are described below.

Regarding the mouse inner ear expression, the first three interesting candidates are *MTMR7*, *R3HDM1,* and *SLC9A9*, characterized by higher expression in HC, with enrichment in cochlear HC. Indeed, previous work by Elkon et al. [30] suggests that genes over-expressed in the HC are good candidates for hearing function, since this expression pattern characterized in their work genes was associated with hearing loss.

*MTMR7* encodes a member of the myotubularin family of tyrosine/dual-specificity phosphatases and was expressed in all the human inner ear tissues assessed, even though at low levels. In our work, this gene was associated with 1 kHz and the association was replicated in both PTAM and PTAH traits, suggesting a possible role of this gene in hearing system although, to date, there are no other associations of this gene with the hearing function.

*R3HDM1* encodes the R3H Domain-Containing Protein, a poorly characterized protein that could have a poly(A) RNA-binding function [32]. Besides being over-expressed in the mouse HC, it showed medium expression in all human inner ear tissues available and in all the cell types of one probe of mouse microarray data, reinforcing its possible role in this system. We found that *R3HDM1* was associated with the 500 Hz trait and this signal was replicated in PTAM.

Another interesting finding involved *SLC9A9*, which encodes a sodium/proton exchanger that is a member of the solute carrier 9 protein family. This protein localizes to the late endosomes and may play an important role in maintaining cation homeostasis [33]. The spectrum of *SLC9A9*-associated diseases in humans includes attention deficit hyperactivity disorder (ADHD), autism spectrum disorders (ASDs), epilepsy, multiple sclerosis, and cancer [34]. To date, there are no clear associations of this gene with the hearing system, even though some studies in animal models suggest it might be involved in the ion homeostasis. In fact, *SLC9A9*, also known as *NHE9*, together with the sodium–hydrogen exchangers *NHE6*, is a member of the solute carrier (SLC) gene superfamily and it has been demonstrated that *Nhe6* knockout mice show significant hearing loss, thus suggesting that NHEs members might play important roles in normal hearing in the mammalian cochlea [35]. According to the data of expression in the human inner ear, *SLC9A9* is detected, even though at low levels. We detected a suggestive association between *SLC9A9* and low frequency traits (250 Hz and PTAL), providing an additional link between this gene and the hearing system.

Six genes, although not over-expressed in mouse HC, had medium or strong expression in all the human inner ear tissues profiled: *BNIP3L, MAP3K20*, *MATN2*, *MYO1E, TGFB2,* and *YTHDC2*.

*BNIP3L* encodes a protein called BCL2/adenovirus E1B 19 kDa protein-interacting protein 3-like (NIP3L/NIX), which is involved in apoptosis and mitochondrial clearance [36]. It has been hypothesized that variants in this gene might be associated with schizophrenia [36]; however, there are no definitive data about the role of this gene in human disease so far. Two independent works on mouse animal models recently revealed a significant decrease in *BNIP3L* expression in the mouse auditory cortex during aging correlating with impaired mitophagy. This occurrence may contribute to the cellular changes observed in an old central auditory system, resulting in age-related hearing loss. Thus, NIP3L/NIX may play a vital role in maintaining cochlear cell homeostasis during the aging process of the hearing system [37,38]. *BNIP3L* was detected at medium levels in all human samples, with strong expression in saccule paralleled by strong expression detected in all mouse cell types. In this light, the suggestive association identified between 8 KHz and rs17402216:A>C (an intronic variant of *BNIP3L*) further supports the possible link between this gene and the hearing system.

*MAP3K20* encodes a serine-threonine kinase that belongs to the MAPKKK family of signal transduction molecules. The protein mediates γ radiation signaling leading to cell cycle arrest and it plays a role in cell cycle checkpoint regulation in cells, in addition to exhibiting pro-apoptotic activity [39]. It has been shown that this gene might have a role in neoplastic cell transformation and cancer development [40], and when mutated causes Centronuclear myopathy 6 with fiber-type disproportion (MIM#:617760), and a syndrome called Split-foot malformation with mesoaxial polydactyly (SFMMP) (MIM# 616890), an autosomal recessive disorder characterized by a split-foot defect, mesoaxial polydactyly, nail abnormalities of the hands, and sensorineural hearing loss. The gene does not seem expressed in the mouse inner ear; however, as previously mentioned, one of the clinical features of SFMMP is also hearing loss. Moreover, the gene was expressed in the human inner ear, and we detected an interesting association between the rs113132813 intronic variant and the PTAM.

*MATN2* encodes a member of the von Willebrand factor A domain containing protein family, which is thought to be involved in the formation of filamentous networks in the extracellular matrices of various tissues. The specific function of this gene has not yet been determined [41]. It looks like it might play a critical role in the differentiation and repair processes of skeletal muscles, peripheral nerves, liver, and skin, however, it has also been implicated in tumor growth or suppression [42]. The gene displayed high expression in the human cochlea, medium expression in the other human ear tissues, and strong expression in all the mouse cell types (probe 4780386). Here, we identified an association between the intronic variant rs11996075 and the 1 KHz, likely suggesting a novel function of this gene.

A suggestive association was identified between 2 kHz and rs67412566:A>T, an intronic variant of *MYO1E*. This gene belongs to the non-muscle class I myosins, a sub-class of the unconventional myosin protein family, one of the earliest associated with hearing loss in humans and mice. Indeed, several members of this family have already been correlated with autosomal dominant and autosomal recessive hearing loss [43]. Regarding *MYO1E*, variants of this gene have been associated with Familial Focal Segmental Glomerulosclerosis (MIM# 614131) [44], and so far, no direct associations with the hearing system have been reported. Nevertheless, *MYO1E* had medium expression in all the human tissues available, a pattern paralleled in the mouse data with medium to strong expression in all cell types. Moreover, expression studies in neonatal rodent auditory and vestibular epithelia confirmed Myo1e expression in the hair cells of both the auditory and vestibular epithelia, in particular in the cuticular plate, being, together with Myo1b and Myo1c, one of the most abundantly expressed genes of the Myo1 family [45]. Together with the association here detected, this evidence provides a possible link between this gene and the hearing function.

An interesting locus, suggestively associated with 8 kHz, is localized in an intronic region belonging to *TGFB2*, a gene that encodes a secreted ligand of the TGF-β superfamily. These ligands can bind different TGF-β receptors, inducing the recruitment and activation of SMAD transcription factors that regulate gene expression. Interestingly, these proteins have been described as involved in the otic capsule development, the formation and survival of spiral ganglion, and indirectly in the cochlear tonotopic organization in mouse models [46]. Variants in this gene have been associated with Loeys–Dietz syndrome 4 (MIM# 614816). According to the literature, the *TGFB2* fully knock-out mouse model exhibited perinatal mortality and a wide range of developmental alterations, including inner ear defects [47]. Furthermore, a work describing the protective effect of another member of the TGFB ligands, *TGFB1*, against noise-induced hearing loss in mouse models, also demonstrated a decrease in *Tgfb2* gene expression after noise exposure [46], supporting the idea that some TGF-β factors can have a potential role in the normal hearing function. 

*YTHDC2* belongs to the DExD/H-box family of ATP-dependent RNA helicases, whose members are involved in RNA processing and metabolism, including transcription, alternative splicing, and degradation [48]. Variants in this gene have been associated with autism spectrum disorder [49] and pancreatic adenocarcinoma [50]; however, there is still no evidence of a certain correlation with human disease. The gene was mostly expressed at low levels in the mouse inner ear while medium expression was observed in all the human samples. Here, we identified the association of the downstream variant rs77567880 and the 8 KHz, likely suggesting a novel function of this gene.

Finally, another two genes deserve attention since they had medium expression in the human cochlea and medium or strong expression in all the cell types in at least one probe of mouse: *ELP5* and *PCNT*.

*ELP5* encodes one of the six subunits of the elongator complex. The elongator is a protein complex involved in multiple processes including transcription regulation, α-tubulin acetylation, and tRNA modification, and its defects have been shown to cause human diseases such as familial dysautonomia [51]. Its mutations are likely involved in migration, invasion, and tumorigenicity of melanoma cells [52], and together with *CLDN7* it has been associated to the modification of adiponectin response to lifestyle intervention in overweight/obese diabetic individuals [53]. To date, data on the possible involvement of this gene in the hearing system are not available. Nevertheless, the gene is expressed in the mouse inner ear, as well as in the human cochlea. Moreover, here we identified an association between the intronic variant rs2106842 and the PTAH and 4 KHz traits, likely suggesting a novel function of this gene.

*PCNT* encodes a large centrosomal coiled-coil protein, the pericentrin, likely involved in brain development [54]. *PCNT* mutations are causative of Microcephalic osteodysplastic primordial dwarfism type 2 (MIM# 210720). The gene is expressed in both mouse and human cochlea, despite its role in the hearing system not being known. Here, we identified a suggestive association between rs11909986 and 1 KHz that could possibly suggest a novel gene function.

The remaining genes (i.e., *CCDC177*, *DLGAP2*, *OOSP1,* and *PTPRN2*), despite the suggestive correlations here identified and the replica in an independent cohort, displayed either weak or no expression in the human inner ear, and, to date, there is no evidence of their contribution to the hearing function. Therefore, additional studies are needed to understand the molecular mechanisms that could link these genes with the hearing system.

The findings reported in this study are in line with the complex nature of the hearing function, in which the accumulation of individual variants with a small effect and environmental factors contribute to the formation of the final phenotype.

Among the 15 detected genes, none were found in previous GWAS/Meta-analyses on hearing function and related traits, although some of the cohorts involved in this study have been already included in previous works [7,8,9,10]. This discrepancy is probably due to differences in sample composition (i.e., some of these cohorts were not included in any previous studies or they were characterized by a different number of samples), different imputation reference panels (for FVG, VBI, and CAR the Italian Genome Reference Panel [11] was used), or different phenotype definitions.

Further experiments and in vivo models will be needed to confirm these results. As an example, gene-specific mouse models could be generated, considering that mice represent the gold standard for the study of genes involved in the hearing system. Nevertheless, for those genes not clearly expressed in the mouse inner ear, other species, such as zebrafish, macaque, or common marmoset, should be also considered.

In addition, a deeper evaluation of the expression pattern of each gene (e.g., expression profiles in each cochlear turn, cell subtypes, etc.) could be useful for better defining their contribution to the hearing function. In fact, it is known that defects in a precise region of the inner ear are related to specific hearing frequencies (i.e., high-frequency sounds lead to maximal basilar membrane movement at the “base” of the cochlea, while low-frequency sounds involve the apical parts of the basilar membrane [55]).

However, considering the multifactorial nature of this trait, other elements (such as the interaction with other genes, environmental factors, etc.) should also be taken into account to fully understand the contribution of the identified genes.

Nonetheless, these findings significantly contribute to understanding the genetic bases underlying the hearing function in the long-term perspective of developing personalized therapeutic approaches, preventive strategies, and diagnostic screenings for hearing impairment.

## Figures and Tables

**Figure 1 genes-12-01228-f001:**
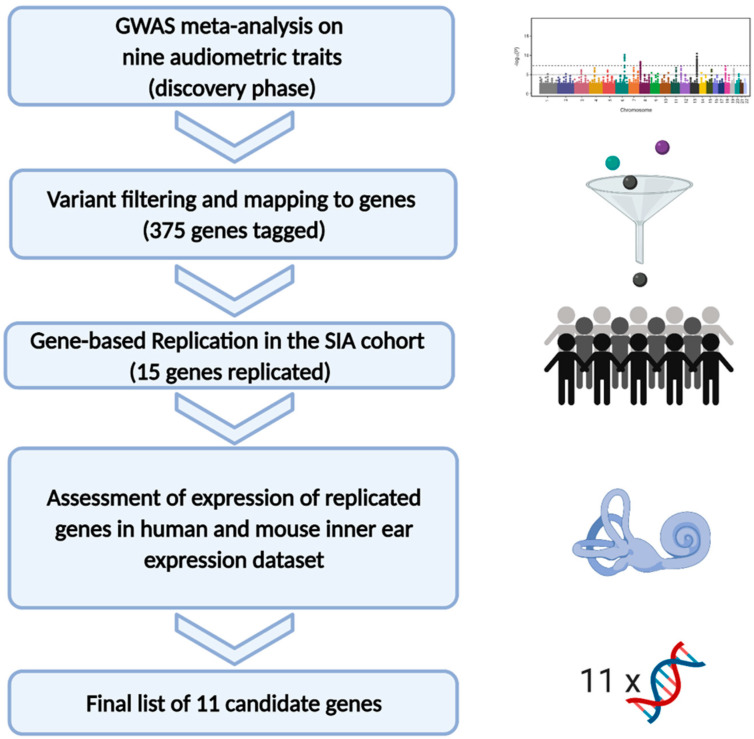
The picture summarizes the workflow applied in the present study.

**Figure 2 genes-12-01228-f002:**
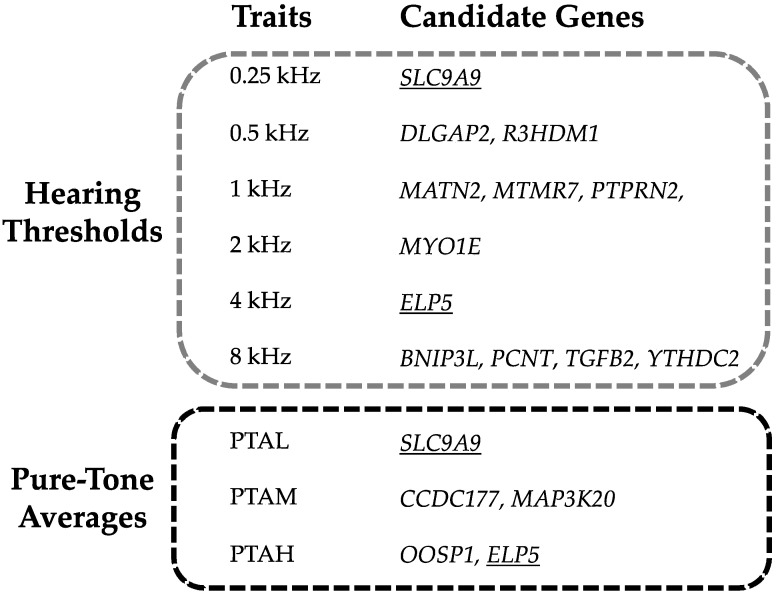
The picture describes the candidate genes for each trait divided in hearing thresholds and Pure-Tone Averages (PTA). The genes found for more than one trait are underlined.

**Figure 3 genes-12-01228-f003:**
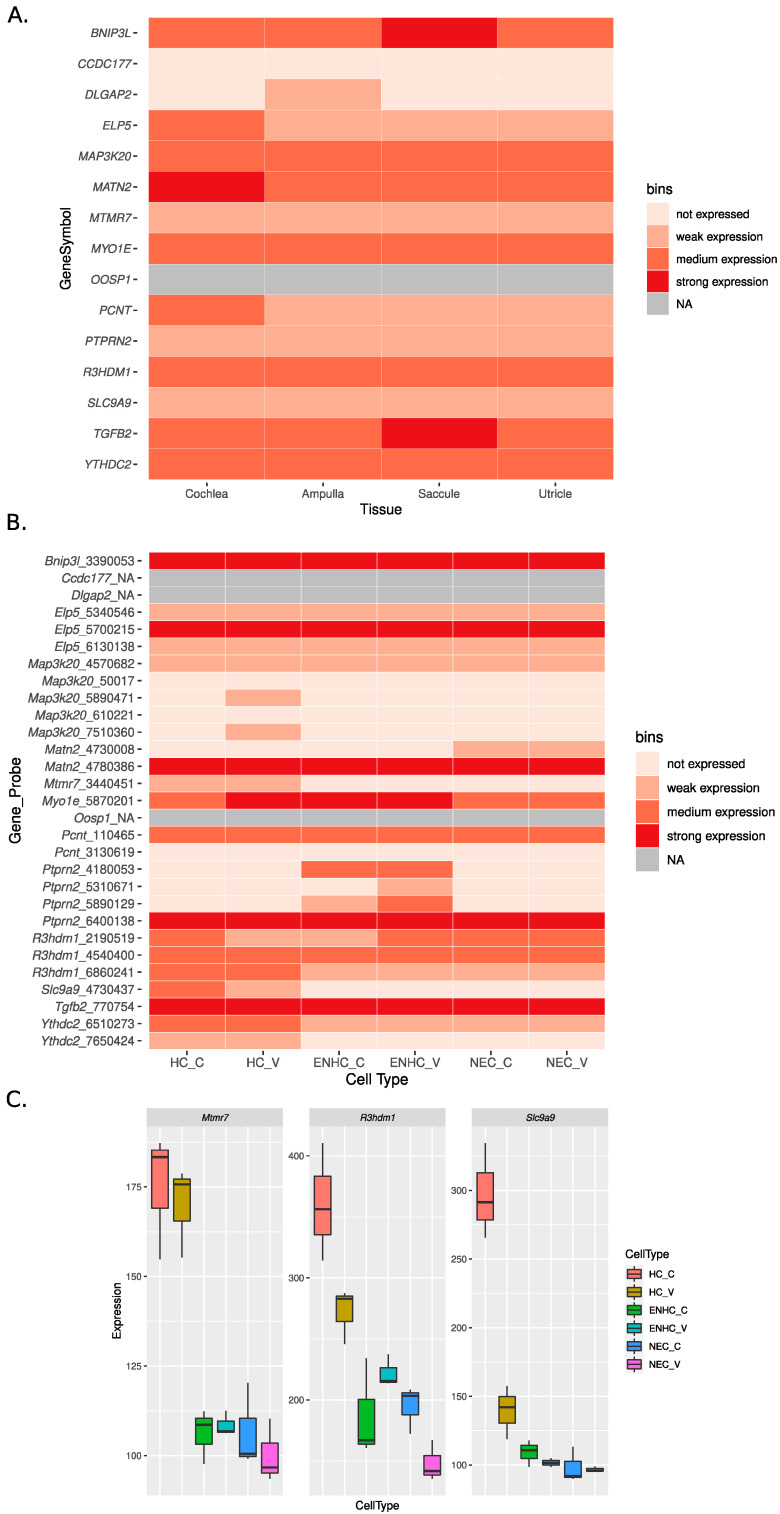
Expression profiles of the replicated genes in human (**A**) and mouse (**B**,**C**). Human cochlea expression represents the mean across the three replicates. In panel (**A**,**B**), for readability, expression was categorized into “expression strength” bins. For human data (panel A, RNA-seq), we defined weak expression for values between 1 and 10 FPKM, medium expression for values between 10 and 50 FPKM, and strong expression for values above 50 FPKM. Genes with values below 1 FPKM were considered as not expressed. For mouse data (panel (**B**), microarray), following the original publication, we defined weak expression for intensities between 125 and 250, medium expression for intensities between 250 and 500, and strong expression for intensities above 500. Genes with intensity below 125 were considered not expressed. NA: data not available. In Figure 3B the mean expression over three replicates is shown. We note that for some of the gene, expression is reported for more than one probe: for clarity we display in Figure 3B intensities for each probe of the corresponding gene. Panel (**C**) shows the expression levels for the three genes with HC enriched expression in mouse.

**Table 1 genes-12-01228-t001:** The table shows the 15 genes identified by discovery and replication analysis. Columns 2–13 are referred to the discovery GWAS meta-analysis performed with METAL, while the last two are referred to the replication step obtained by gene-based analysis using MAGMA. The columns are Gene: candidate gene; Trait: phenotype in which the gene was identified; SNP: most significant SNP; Chr: chromosome; Position: position of the variant (build hg19); Effect: variant function as obtained using VEP; Alleles: effect allele and other alleles in GWAS meta-analysis; Freq: frequency of the effect allele; N: total analyzed sample size for the variant in the given meta-analysis; Beta effect from GWAS meta-analysis; StdErr: standard error of the beta; *p*-value: *p*-value from GWAS meta-analysis; Direction: string summarizing the effect directions (positive or negative) in the involved cohorts, presented in this order: FVG, VBI, CAR, SR, AWP, RS1, RS2, RS3, FHS. A question mark denotes that the variant was not available for the specific cohort; *p*-value PTAM and *p*-value PTAH: *p*-value obtained in the replication step for the gene indicated in the first column in PTAM (Pure Tone Average of thresholds at Medium frequencies) and PTAH (Pure Tone Average of thresholds at High frequencies) phenotypes, respectively. The last two columns indicate in bold the significant (<0.05) *p*-value.

Discovery GWAS Meta-Analysis													Replication Analysis	
Gene	Trait	SNP	Chr	Position	Effect	Alleles	Freq	N	Beta	StdErr	*p*-Value	Direction	*p*-Value PTAM	*p*-Value PTAH
*BNIP3L*	8 kHz	rs17402216	8	26325225	intron	A/C	0.944	9003	0.043	0.008	4.69 × 10^−7^	++-+-++++	**5.99 × 10^−3^**	**2.56 × 10^−3^**
*CCDC177*	PTAM	rs2025133	14	70058292	upstream gene	A/G	0.829	8989	−0.021	0.005	3.87 × 10^−6^	---------	2.49 × 10^−1^	**1.38 × 10^−2^**
*DLGAP2*	500 Hz	rs181305232	8	1539701	intron	T/C	0.034	6152	−0.082	0.018	5.68 × 10^−6^	---??---?	7.03 × 10^−1^	**4.57 × 10^−3^**
*ELP5*	4 kHz	rs2106842	17	7162451	intron	A/G	0.646	5729	−0.022	0.005	2.84 × 10^−6^	--++---??	**3.58 × 10^−2^**	6.69 × 10^−1^
	PTAH							5729	−0.015	0.003	6.23 × 10^−6^	--+----??		
*MAP3K20*	PTAM	rs113132813	2	173956255	intron	A/G	0.021	5801	−0.075	0.016	1.49 × 10^−6^	?-+??----	**4.42 × 10^−2^**	6.97 × 10^−1^
*MATN2*	1 kHz	rs11996075	8	98910287	intron	T/C	0.118	7932	0.034	0.007	4.83 × 10^−6^	++++?+++-	1.28 × 10^−1^	**4.22 × 10^−2^**
*MTMR7*	1 kHz	rs12155974	8	17299124	upstream gene	T/C	0.911	7932	−0.050	0.010	2.35 × 10^−7^	----?----	**7.04 × 10^−3^**	**8.09 × 10^−4^**
*MYO1E*	2 kHz	rs67412566	15	59523608	intron	A/T	0.833	6820	0.048	0.011	7.66 × 10^−6^	++++?+++?	**4.19 × 10^−2^**	9.20 × 10^−1^
*OOSP1*	PTAH	rs111389524	11	59757661	downstream gene	A/G	0.036	7895	−0.044	0.010	7.14 × 10^−6^	----?----	**4.21 × 10^−4^**	**1.55 × 10^−2^**
*PCNT*	8 kHz	rs11909986	21	47780161	intron	A/G	0.332	8348	−0.018	0.004	7.86 × 10^−6^	---?-----	1.87 × 10^−1^	**3.02 × 10^−2^**
*PTPRN2*	1 kHz	rs11514653	7	158356977	intron	C/G	0.022	6954	0.091	0.019	2.78 × 10^−6^	++??++++?	4.20 × 10^−1^	**4.78 × 10^−2^**
*R3HDM1*	500 Hz	rs7560535	2	136412472	intron	A/G	0.175	6431	0.032	0.007	7.39 × 10^−6^	?+++?++++	**9.25 × 10^−3^**	2.41 × 10^−1^
*SLC9A9*	250 Hz	rs76168782	3	143270684	intron	T/G	0.946	7244	−0.054	0.012	6.86 × 10^−6^	---??----	**2.47 × 10^−3^**	8.60 × 10^−1^
	PTAL							7264	−0.046	0.010	3.27 × 10^−6^	---??----		
*TGFB2*	8 kHz	rs149269977	1	218546474	intron	T/C	0.026	7874	−0.071	0.016	7.86 × 10^−6^	---+----?	**2.50 × 10^−2^**	4.06 × 10^−1^
*YTHDC2*	8 kHz	rs77567880	5	113145730	downstream gene	A/G	0.015	6055	−0.102	0.022	4.36 × 10^−6^	??--?--+-	3.14 × 10^−1^	**4.84 × 10^−2^**

## Data Availability

FVG-VBI-CAR: a subset of the data is already available on the European Genome-phenome Archive (EGA) at the following links: FVG cohort: BAM files https://www.ebi.ac.uk/ega/studies/EGAS00001000252, accessed on 7 May 2021; sample list, vcf files https://www.ebi.ac.uk/ega/studies/EGAS00001001597, accessed on 7 May 2021; https://www.ebi.ac.uk/ega/datasets/EGAD00001002729, accessed on 7 May 2021; VBI cohort: BAM files https://www.ebi.ac.uk/ega/studies/EGAS00001000398, accessed on 7 May 2021; https://www.ebi.ac.uk/ega/studies/EGAS00001000458, accessed on 7 May 2021; CAR cohort: BAM files https://www.ebi.ac.uk/ega/studies/EGAS00001000460, accessed on 7 May 2021. A vcf file including all the INGI variants (SNPs and INDELs) with information on allele frequencies in the whole dataset and each cohort has been submitted to the European Variation Archive (EVA) study accession number: PRJEB33648. The data is accessible at the following link: https://www.ebi.ac.uk/ena/data/view/PRJEB33648, accessed on 7 May 2021. Hearing data for Framingham Heart study participants can be requested by applying to the Database of Genotype and Phenotype (dbGaP, https://dbgap.ncbi.nlm.nih.gov/aa/wga.cgi?page=login, accessed on 7 May 2021).

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
