# Peer review of "Hearing Function: Identification of New Candidate Genes Further Explaining the Complexity of This Sensory Ability"

_genes, 2021, doi:10.3390/genes12081228_

Round 1

Reviewer 1 Report

In this manuscript, Concas et al. reported their results of GWAS meta-analysis about hearing loss. The construction of the manuscript is excellent. However, I hope that they add a schema and a discussion to the manuscript.

Minor points)

  1. They mentioned the relationships between the several candidate genes and hearing thresholds in a particular frequency (e.g. MTMR7 and 1kHz). I hope that the authors add a schema indicating these relationships at a glance. I believe that it would be helpful for the readers.

  1. In line 376, they mentioned the importance of the in vivo model in the future. However, in the several candidate genes, there are inter-species differences in the expression patterns (e.g., YTHDC2 written in Line 343). These might indicate that a mice model does not help investigate the function of these genes. I hope that the authors add a discussion about any possible alternative approach to this point. I believe that it will be valuable for the future approach. In my opinion, the primate model animals including macaque or common marmoset might be helpful.

Author Response

In this manuscript, Concas et al. reported their results of GWAS meta-analysis about hearing loss. The construction of the manuscript is excellent. However, I hope that they add a schema and a discussion to the manuscript.

Response: We thank the Reviewer for the appreciation of our work.

Minor points

  1. They mentioned the relationships between the several candidate genes and hearing thresholds in a particular frequency (e.g. MTMR7 and 1kHz). I hope that the authors add a schema indicating these relationships at a glance. I believe that it would be helpful for the readers.

Response: We thank the Reviewer for the comment. We add a scheme (Figure 2 in the text) as required.

  1. In line 376, they mentioned the importance of the in vivo model in the future. However, in the several candidate genes, there are inter-species differences in the expression patterns (e.g., YTHDC2 written in Line 343). These might indicate that a mice model does not help investigate the function of these genes. I hope that the authors add a discussion about any possible alternative approach to this point. I believe that it will be valuable for the future approach. In my opinion, the primate model animals including macaque or common marmoset might be helpful.

Response: We thank the Reviewer for the comment. As the Reviewer pointed out, some of the candidate genes we described display differences in the human and mouse tissue expression patterns. However, we think this does not imply that the mice model could not help investigate the function of these genes. In fact, within the hearing loss field, many research studies have been carried out highlighting the successes of the use of mice in hearing and vestibular phenotypes. This is related to the fact that the mouse inner ear follows standard mammalian principles (e.g., cell types, structures and molecular mechanism) resembling human hearing system. Moreover, mice are worldwide used for the study of different disorders. For this reason, many of the available databases often contain a lot of information relative to this species, rather than the others, facilitating the experimental procedures. Finally, the expression data we checked is relative to a specific time point (i.e., post-natal day 1), thus it would be worth testing the expression levels also at additional time points (both at embryonic and post-natal stages), to assess gene expression during time. Nevertheless, we agree with the Reviewer that other species (including macaque or common marmoset) or other technologies could be used whenever the mouse model wouldn’t be suitable for such experiments. We added some comments in the discussion.

Reviewer 2 Report

Comments for Authors

This manuscript is devoted to the search for candidate genes involved in hearing function by the comprehensive GWAS meta-analysis using genotype and audiometric data collected in nine large cohorts (~ 9,000 subjects). 15 out of 375 genes were replicated in the independent cohort (Italian Salus in Apulia cohort, n=1,774). Based on the expression patterns in humans and mouse, 11 genes were proposed as potential candidates associated with one or more hearing phenotypes.

My comments:

The authors report 11 (15) new genes revealed by the GWAS meta-analysis potentially involved in modulation of auditory function. This study certainly contributes to the knowledge of the genetic determinants of hearing function and related diseases. However, I believe that the Introduction should more fully present previous studies on this issue, at least, the study apparently carried out by the same research team - Nagtegaal et al. “Genome-wide association meta-analysis identifies five novel loci for age-related hearing impairment.” Sci Rep. 2019 Oct 23; 9 (1): 15192. doi: 10.1038 / s41598-019-51630-x. In addition, in the Discussion, the authors should compare the results of this study with previously obtained data and somehow discuss them. My question: Why the spectrum of genes, identified in this study, does not coincide at all with those that were identified in the study by Nagtegaal et al., where, apparently, almost the same cohorts, phenotypic traits, and methodological approaches were used? Please, explain this issue.

Minor comment: Figure 2: the names of genes (human, mouse) must be in italic.

Author Response

This manuscript is devoted to the search for candidate genes involved in hearing function by the comprehensive GWAS meta-analysis using genotype and audiometric data collected in nine large cohorts (~ 9,000 subjects). 15 out of 375 genes were replicated in the independent cohort (Italian Salus in Apulia cohort, n=1,774). Based on the expression patterns in humans and mouse, 11 genes were proposed as potential candidates associated with one or more hearing phenotypes.

My comments:

The authors report 11 (15) new genes revealed by the GWAS meta-analysis potentially involved in modulation of auditory function. This study certainly contributes to the knowledge of the genetic determinants of hearing function and related diseases. However, I believe that the Introduction should more fully present previous studies on this issue, at least, the study apparently carried out by the same research team - Nagtegaal et al. “Genome-wide association meta-analysis identifies five novel loci for age-related hearing impairment.” Sci Rep. 2019 Oct 23; 9 (1): 15192. doi: 10.1038 / s41598-019-51630-x. In addition, in the Discussion, the authors should compare the results of this study with previously obtained data and somehow discuss them.

Response: We thank the Reviewer for the comment. As kindly required, we added in the Introduction section a sentence to mention the previous work of our co-author Dr. Nagtegaal. We also added a sentence in the Discussion section to explain the difference in gene discovery between present work and our past published findings.

My question: Why the spectrum of genes, identified in this study, does not coincide at all with those that were identified in the study by Nagtegaal et al., where, apparently, almost the same cohorts, phenotypic traits, and methodological approaches were used? Please, explain this issue.

Response: As written in the text, there are many differences between this work and the previous one. In particular, any discrepancy is probably due to differences in samples composition (i.e., some of these cohorts were not included in any previous studies or they were characterized by a different number of samples), different imputation reference panels (for FVG, VBI and CAR the Italian Genome Reference Panel (https://doi.org/10.1038/s41431-019-0551-x)), or different phenotypes definition. For all these reasons, each GWAS is an independent study that can reveal new or different results. Thus, data replication in independent cohort is an essential step.

Minor comment: Figure 2: the names of genes (human, mouse) must be in italic.

Response: As required, we edited the genes’ names in italic.
